# The added diagnostic value of RT-PCR on faeces for the diagnosis of COVID-19

Nathalie Van der Moeren[1,2,3*], Rik van den Biggelaar[1,2,3], Karin B. Gast[4],
Jaco J. Verweij[1,2,3], Barbara J.M. Bergmans[5], Joep J. J.M. Stohr[1,2,3], Jean-Luc Murk[1,2,3]

1 Department of Medical Microbiology and Immunology, Elisabeth-TweeSteden Hospital, Tilburg, the Netherlands, 2 Department of Medical Microbiology and Immunology, Amphia Hospital, Breda, the Netherlands, 3 Microvida, Laboratory of Medical Microbiology and Immunology, Elisabeth-TweeSteden Hospital, Tilburg, the Netherlands, 4 Department of Medical Microbiology, Eurofins PAMM, Veldhoven, the Netherlands, 5 Laboratory for Microbiology Twente Achterhoek (Labmicta), Hengelo, the Netherlands

* n.vdmoeren@gmail.com

## Abstract

### Objectives

To determine whether combining a SARS-CoV-2 RT-PCR on a fecal sample [FS] with a RT-PCR on an upper respiratory tract sample [URTS] results in additional COVID-19 diagnoses.

### Methods

We conducted a retrospective observational study at a regional hospital in The Netherlands from 27 February 2020–30 June 2020. Patients presenting with COVID-19-like symptoms for who a SARS-CoV-2 RT-PCR on both URTS and FS were obtained within 24 hours were included. We calculated the difference in positive RT-PCR when combining URTS/FS compared to URTS alone, overall and stratified by symptom duration and disease severity.

### Results

Three hundred eighty-six patients were included of which 63 had a positive RT-PCR on URTS [n = 8], FS [n = 19] or both [n = 36], corresponding to a prevalence of 16.3%. The addition of testing FS increased the number of COVID-19 diagnoses by 31.8% [95%CI 20,3%-43,2%].

### Conclusions

We showed that adding SARS-CoV-2 RT-PCR on FS to URTS yields significantly more COVID-19 diagnoses. The inclusion of an FS may therefore be considered in patients with a negative URTS and high suspicion of COVID-19.

**Data availability statement:** All relevant data are within the paper and its Supporting information files.

**Funding:** The author(s) received no specific funding for this work.

**Competing interests:** The authors have declared that no competing interests exist.

## Introduction

The primary method to diagnose COVID-19 is RT-PCR on an upper respiratory tract sample [URTS]. The clinical sensitivity of this method is estimated to be 80%-95%. Several factors such as sampling technique, time since symptom onset and disease severity influence the test sensitivity [1–4]. Very early on in the pandemic, it became clear that a single URTS was regularly insufficient to diagnose COVID-19. Several methods were used to improve diagnostic certainty: repeated testing, obtaining samples from the lower respiratory tract and performing CT scans to look for characteristic lung abnormalities [5–7].

Several studies have shown that SARS-CoV-2 RNA can also be detected in faecal samples [FS] in a proportion of patients with a positive URTS RT-PCR [8–10]. The evidence on the potential added value of a SARS-CoV-2 faecal RT-PCR to improve diagnostic certainty is however scarce and to our knowledge no systematic studies have been performed [3,11].

Hence the present study in which we aim to estimate the complementary role of SARS-CoV-2 RT-PCR on an FS for the diagnosis of COVID-19 using historical records of our hospital.

## Methods

### Objectives

The primary objective was to determine whether adding a SARS-CoV-2 RT-PCR on a FS to a RT-PCR on a URTS increases the number of laboratory-confirmed COVID-19 diagnoses, overall and in relation to symptom duration and disease severity.

### Study design

Retrospective observational study.

### Setting

The study was performed at the Elisabeth-TweeSteden Hospital Tilburg, the Netherlands, from 27 February 2020–30 June 2020.

Confronted with the shortcomings of SARS-CoV-2 testing on a single URTS and based on the literature on the detection of SARS-CoV-2 RNA in faeces, faecal testing was added to the hospital's diagnostic algorithm for patients with suspected COVID-19. FS could not always be obtained as patients were not always able to defecate in time. Furthermore, a number of physicians resorted to two-tier testing: awaiting results from the RT-PCR on URTS and adding RT-PCR on FS when clinical suspicion for COVID-19 remained high after a negative URTS result.

### Patient selection and sample collection

Patients that [1] presented at the hospital within the study period [2], had a cough and/or dyspnoea and/or fever without apparent cause and [3] for whom SARS-CoV-2 RT-PCR results were available on both a URTS [nose/throat/combined nose-throat/

nasopharyngeal sample] and an FS obtained within the same 24 hours were included. When patients had been included in the study in the 14 days before, they were not included a second time.

Test results from all included patients, were extracted from the laboratory information system on 24 November 2023. The time between symptom onset and sampling and the disease severity were manually retrieved from the electronic patient files. According to the WHO definitions, the patients' COVID-19 severity was classified as non-severe, severe, or critical. After this classification, all data was anonymised [12].

## SARS-CoV-2 RT-PCR

An in-house duplex RT-PCR for SARS-CoV-2 E-gen and PDV was performed. Total nucleic acids [NA] were extracted using the QIAsymphony DSP virus/pathogen midi kit and pathogen complex 400 protocol of the QIAsymphony Sample Processing [SP] system [Qiagen, Hilden, Germany]. In each sample, a fixed amount of phocine distemper virus [PDV] was added within the isolation lysis buffer to serve as an internal control for the isolation procedure and to monitor the inhibition of the real-time PCR. A duplex PCR for SARS-CoV-2 E-gen/PDV was performed with optimised primer and probe concentrations in a volume of 25 μL with TaqMan® Fast Virus 1-Step Master Mix [Thermofisher, Waltham, United States] and 10 μL NA sample.[7,8] Amplification consisted of 5 minutes at 50°C, 15 minutes at 95°C, followed by 45 cycles of 15 seconds at 95°C, 30 seconds at 60°C, and 15 seconds at 72°C. Amplification, detection, and analysis were performed with the Rotor-gene real-time detection system [Qiagen]. Negative and positive control samples were included in each amplification run.

Faecal samples were pre-treated: 250 mg of faeces was suspended in 800 μl STAR buffer [Roche, Basel, Switzerland]. After centrifugation, RT-PCR was performed on the supernatant using the abovementioned method. RT-PCR procedure validation was performed according to International Standards Organization guidelines 15189 [http://www.iso.org/iso/search.htm].

### Analysis

The numbers of positive SARS-CoV-2 RT-PCR on URTS and FS separately were compared to the number of positive SARS-CoV-2 RT-PCR on URTS/FS combined, overall and stratified by disease severity [non-severe, severe, critical] and time since symptom onset [0–7 days, 8–14 days and > 14 days]. Subsequently, the difference in positive RT-PCR when combining URTS/FS compared to URTS alone was calculated.

### Statistical analysis

Exact Clopper-Pearson confidence intervals were calculated for the proportion of RT-PCR positive on URTS and FS compared to URTS/FS combined. The McNemar test was used to compute confidence intervals and p-values for the difference in positive RT-PCR between combined URTS/FS and URTS alone. All statistical analyses were performed with SPSS 29.0.1.0 and Medcalc 22.016. Results with a p-value < 0.05 were considered significant.

### Ethics statement

The study protocol was submitted to the medical ethical board 'Medical Research Ethics Committees Brabant' and was granted an exemption from the Dutch Medical Scientific Research Act (WMO).

As the study was retrospective and the primary objective of faecal testing was to optimise the local diagnostic strategy, written informed consent was not deemed necessary. The study's planning, conduct and reporting were in accordance with the Declaration of Helsinki, as revised in 2013.

## Results

During the study period, 3480 URTS and 654 FS SARS-CoV-2 RT-PCR were performed from patients presenting at the hospital, positivity rates were respectively 18.4% [n = 641] and 12.4% [n = 81]. Three hundred eighty-six patients met the

inclusion criteria, of which 63 had a positive RT-PCR on URTS [n = 8], FS [n = 19] or both [n = 35], corresponding to a prevalence of 16.3% [Table 1, Fig 1]. The mean age was 68, and the female-to-male ratio was 42.5% [n = 164].

Overall, the addition of testing FS increased the number of COVID-19 diagnoses by 31.8% [95%CI 20,3%-43,2%]. Likewise, a significantly higher diagnostic yield was observed in patients with non-severe and severe illness and in samples obtained during the first and second week after symptom onset [p < 0.05]. A positive trend was observed in patients with critical illness and when samples were obtained >14 days after symptom onset [p = 0.0625]. Overall, the number of RT-PCR positive cases detected by a URTS and FS were 68.3% and 87.3% respectively compared to the number detected by URTS/FS combined [Table 2].

## Discussion

We found that adding a SARS-CoV-2 RT-PCR on an FS to a RT-PCR on a URTS significantly increased the number of laboratory-confirmed COVID-19 diagnoses [p < 0,001].

Literature on the diagnostic value of adding SARS-CoV-2 RT-PCR on an FS to standard testing on a URTS is scarce. Bergmans et al. showed that an RT-PCR on an FS detected 39.6% extra SARS-CoV-2 positives in patients with an initial negative RT-PCR compared to only 11.2% with an additional URTS RT-PCR [3]. Szymczak et al. evaluated two commercial faecal SARS-CoV-2 RT-PCR assays and detected two COVID-19 cases amongst 29 patients who tested URTS RT-PCR negative [11]. To our knowledge, this is the first study describing the added diagnostic value of RT-PCR on an FS to diagnose COVID-19.

One of the limitations of the study is the probable selection bias of patients with an initial negative RT-PCR on a URTS. In order to reduce the bias of FS testing depending on a negative RT-PCR on a URTS, only patients for which both URTS and FS were tested within 24 hours were included. However, as the percentage of SARS-CoV-2 RT-PCR-positive patients

**Table 1. Crosstabulation of SARS-CoV-2 RT-PCR on upper respiratory samples [URS] and fecal samples [FS].**

|  |  | FS |  |  |
|---|---|---|---|---|
|  |  | Positive | Negative | TOTAL |
| URS | Positive | 35 | 8 | 43 |
|  | Negative | 19 | 323 | 342 |
|  | Failed | 1 | 0 | 1 |
|  | TOTAL | 55 | 331 | 386 |

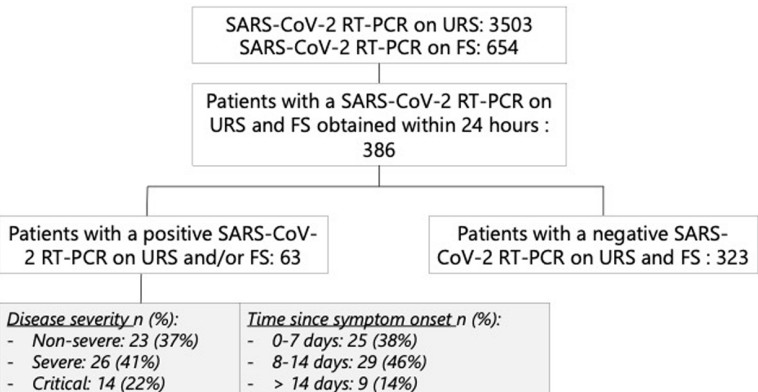

**Fig 1. Overview of included individuals, disease severity and time since symptom onset.**

**Table 2. Diagnostic yield of SARS-CoV-2 RT-PCR on upper respiratory samples (URS) and Fecal samples (FS) compared to SARS-CoV-2 RT-PCR on URS/FS combined, overall and stratified by disease severity and time since symptom onset.**

| | | | | | Disease severity | | | | | | Time since symptom onset | | | |
|---|---|---|---|---|---|---|---|---|---|---|---|---|---|---|
| | Overall (n=63) | | Non-severe (n=23) | | Severe (n=26) | | Critical (n=14) | | 0-7 days (n=25) | | 8-14 days (n=29) | | > 14 days (n=9) | |
| | n | Sens % (95% CI) | n | Sens % (95% CI) | n | Sens % (95% CI) | n | Sens % (95% CI) | n | Sens % (95% CI) | n | Sens % (95% CI) | n | Sens % (95% CI) |
| URS* | 43 | 68,3% (55,3%-79,4%) | 17 | 73,9% (51,6%-89,8%) | 17 | 65,4% (44,3%-82,8%) | 9 | 64,3% (35,1%-87,2%) | 18 | 72% (50,6%-87,9%) | 21 | 72,4% (52,7%-87,3%) | 4 | 44,4% (13,7%-78,8%) |
| FS* | 55 | 87,3% (76,5%-94,4%) | 23 | 100% (85,2%-100%) | 20 | 76,9% (56,4%-91,0%) | 12 | 85,7% (57,2%-98,2%) | 22 | 88% (68,8%-97,4%) | 24 | 82,8% (64,2% - 94,1%) | 9 | 100% (66,4%-100%) |
| URS/ FS >URS** | 20 | 31,8% (20,3%-43,2%) | 6 | 26,1% (8,1%-44,1%) | 9 | 34,6% (16,3%-52,9%) | 5 | 35,7% (10,6%-60,8%) | 7 | 28,0% (10,4%-45,6%) | 8 | 27,6% (11,3%-43,9%) | 5 | 55,6% (23,1%-88,0%) |
| P-value | p<0,0001 | | p=0,0313 | | p=0,0039 | | p=0,0625 | | p=0,0156 | | p=0,0078 | | p=0,0625 | |

detected through RT-PCR on URTS in our study is lower than expected based on the existing literature, a residual bias is likely [3,11]. Furthermore, a relatively limited number of patients was included.

We did not have information about gastrointestinal symptoms in the included individuals; therefore, an overrepresentation of patients with gastrointestinal symptoms cannot be excluded. Furthermore, the study was performed when wild-type SARS-CoV-2 was circulating in mostly immunologic naïve patients, considerably differing from the current viral and immunological conditions. Finally, although the pre-analytic steps to prepare feces for RT-PCR take little extra time and are simple to perform, they would have to be newly implemented in laboratories not already routinely performing RT-PCR on feces (*Clostridioides difficile*, gastro-intestinal viruses a.o.).

The morbidity and mortality of COVID-19 have strongly decreased due to naturally acquired immunity and widespread vaccination [13,14]. Nevertheless, immunocompromised individuals, the elderly and those with a contra-indication for vaccination, remain at risk for severe disease today [15–17]. Furthermore, SARS-CoV-2 will likely be circulating as an endemic virus in the foreseeable future with new variants escaping the existing immunity, lurking [15,16]. Finally, the currently available therapeutics are most effective when administered early on in the disease course [18–20]. Consequently, a correct and prompt diagnosis of a SARS-CoV-2 infection remains important.

In conclusion, we showed that adding SARS-CoV-2 RT-PCR on FS to URTS yields significantly more COVID-19 diagnoses. The inclusion of an FS may therefore be considered in patients with a negative URTS and high suspicion of COVID-19. Further research is warranted as this study was performed retrospectively and may not apply to current viral and immunological conditions.

## Supporting information

**S1 Data. PONE-D-24-25703_Data File.**
(XLSX)

## Author contributions

**Conceptualization:** Nathalie Van der Moeren, Rik van den Biggelaar, Joep J. J.M. Stohr, Jean-Luc Murk.

**Data curation:** Nathalie Van der Moeren, Rik van den Biggelaar, Karin B. Gast, Jaco J. Verweij, Barbara J.M. Bergmans.

**Formal analysis:** Nathalie Van der Moeren, Rik van den Biggelaar.

**Methodology:** Nathalie Van der Moeren, Jaco J. Verweij, Joep J. J.M. Stohr, Jean-Luc Murk.

**Supervision:** Joep J. J.M. Stohr, Jean-Luc Murk.

**Visualization:** Nathalie Van der Moeren.

**Writing – original draft:** Nathalie Van der Moeren.

**Writing – review & editing:** Nathalie Van der Moeren, Rik van den Biggelaar, Karin B. Gast, Jaco J. Verweij, Barbara J.M. Bergmans, Joep J. J.M. Stohr, Jean-Luc Murk.

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
