## [Decision Letter · Decision Letter 0]

23 Dec 2024

PONE-D-24-25703The added diagnostic value of RT-PCR on faeces for the diagnosis of COVID-19PLOS ONE

Dear Dr. Van der Moeren,

Thank you for submitting your manuscript to PLOS ONE. After careful consideration, we feel that it has merit but does not fully meet PLOS ONE’s publication criteria as it currently stands. Therefore, we invite you to submit a revised version of the manuscript that addresses the points raised during the review process.

We look forward to receiving your revised manuscript.

Kind regards,

Vishwanatha R. A. P. Reddy

Academic Editor

PLOS ONE

Journal Requirements:

Reviewers' comments:

Reviewer's Responses to Questions

**Comments to the Author**

1. Is the manuscript technically sound, and do the data support the conclusions?

Reviewer #1: Yes

Reviewer #2: Yes

2. Has the statistical analysis been performed appropriately and rigorously? 

Reviewer #1: I Don't Know

Reviewer #2: Yes

3. Have the authors made all data underlying the findings in their manuscript fully available?

Reviewer #1: Yes

Reviewer #2: Yes

4. Is the manuscript presented in an intelligible fashion and written in standard English?

Reviewer #1: Yes

Reviewer #2: Yes

5. Review Comments to the Author

Reviewer #1: This study showed that combining SARS-CoV-2 RT-PCR testing on fecal samples (FS) with upper respiratory tract samples (URTS) increased COVID-19 diagnoses by 31.8%. Hence, the authors claim that adding FS testing may enhance detection in patients with negative URTS results but high clinical suspicion of COVID-19.

The paper is well-written and deserves consideration. However minor revision is required:

-Table 1, detailing patients' characteristics, is missing.

-Have the authors used a validated assay to perform RT-PCR testing on fecal samples?

-The authors should expand the discussion regarding: the capacity of standard laboratories to perform RT-PCR on fecal samples and the potential delayed positivity of RT-PCR on fecal samples compared to upper respiratory tract testing.

Reviewer #2: The authors presented results that confirmed that use of RT_PCR on fecal samples improves overall diagnostic accuracy. However, this manuscript should qualify as a short communication going by the small dataset used to arrive at this conclusion. Also, the statement provided under the objective appears ambiguous. Authors should make it clearer.

6. PLOS authors have the option to publish the peer review history of their article (what does this mean? ). If published, this will include your full peer review and any attached files.

**Do you want your identity to be public for this peer review?** For information about this choice, including consent withdrawal, please see our Privacy Policy .

Reviewer #1: No

Reviewer #2: **Yes: ** Ayodele Oluwaseun Ajayi

---

## [Author Response · Author response to Decision Letter 1]

21 Mar 2025

Dear Sir, Madam,

We would like to thank you for the opportunity to resubmit a revised copy of the manuscript “The added diagnostic value of RT-PCR on faeces for the diagnosis of COVID-19”.

We greatly appreciate the careful review and adapted the manuscript according to the comments. We believe the manuscript is substantially improved after making the suggested edits.

We hope that these revisions are sufficient to make our manuscript suitable for publication in ‘Plos One’.

Yours sincerely,

Nathalie Van der Moeren

In name of the co-authors

Reviewer 1

This study showed that combining SARS-CoV-2 RT-PCR testing on fecal samples (FS) with upper respiratory tract samples (URTS) increased COVID-19 diagnoses by 31.8%. Hence, the authors claim that adding FS testing may enhance detection in patients with negative URTS results but high clinical suspicion of COVID-19.

The paper is well-written and deserves consideration. However minor revision is required:

- Table 1, detailing patients' characteristics, is missing.

Thank you for this valuable remark. Table 1 is the crosstabulation of SARS-CoV-2 RT-PCR on upper respiratory samples and fecal samples - to which we refer in the text in front of the reference to table 1. As we only had access to age and sex of the included patients, we chose to briefly describe the main patient characteristics (age and sex) in the text (line 144).

- Have the authors used a validated assay to perform RT-PCR testing on fecal samples?

Thank you for this relevant question. The PCR procedure was validated according to the ISO guidelines. We moved the sentence in which we describe this further down in the text to clarify this. (line 117-119)

- The authors should expand the discussion regarding: the capacity of standard laboratories to perform RT-PCR on fecal samples and the potential delayed positivity of RT-PCR on fecal samples compared to upper respiratory tract testing.

Thank you for this justly remark. We added the sentence ‘Finally, although the pre-analytic steps to prepare feces for RT-PCR take little extra time and are simple to perform, they would have to be newly implemented in laboratories not already routinely performing RT-PCR on feces (Clostridioides difficile, gastro-intestinal viruses a.o.).’ to the discussion. (Line 208-211)

Reviewer 2

The authors presented results that confirmed that use of RT_PCR on fecal samples improves overall diagnostic accuracy.

- However, this manuscript should qualify as a short communication going by the small dataset used to arrive at this conclusion. Also, the statement provided under the objective appears ambiguous. Authors should make it clearer.

Thank you for these valuable comment. We recognise the relatively limited number of patients included and added this as a limitation to the discussion section. (line 200)

- Also, the statement provided under the objective appears ambiguous. Authors should make it clearer.

Thank you for this justified remark, we specified the objective. (line 67-69)

---

## [Editor Report · Decision Letter 1]

24 Mar 2025

The added diagnostic value of RT-PCR on faeces for the diagnosis of COVID-19

PONE-D-24-25703R1

Dear Dr. Van der Moeren,

We’re pleased to inform you that your manuscript has been judged scientifically suitable for publication and will be formally accepted for publication once it meets all outstanding technical requirements.

Kind regards,

Vishwanatha R. A. P. Reddy

Academic Editor

PLOS ONE

---

## [Editor Report · Acceptance letter]

PONE-D-24-25703R1

PLOS ONE

Dear Dr. Van der Moeren,

I'm pleased to inform you that your manuscript has been deemed suitable for publication in PLOS ONE. Congratulations! Your manuscript is now being handed over to our production team.

Kind regards,

on behalf of

Dr. Vishwanatha R. A. P. Reddy

Academic Editor

PLOS ONE